# A Loss Function for Generative Neural Networks Based on Watson's Perceptual Model

**Steffen Czolbe**[1]     **Oswin Krause**[1]     **Ingemar Cox**[1,2]     **Christian Igel**[1]

[1]University of Copenhagen, Department of Computer Science
[2]University College London, Department of Computer Science
`{per.sc,oswin.krause,ingemar.cox,igel}@di.ku.dk`

## Abstract

To train Variational Autoencoders (VAEs) to generate realistic imagery requires a loss function that reflects human perception of image similarity. We propose such a loss function based on Watson's perceptual model, which computes a weighted distance in frequency space and accounts for luminance and contrast masking. We extend the model to color images, increase its robustness to translation by using the Fourier Transform, remove artifacts due to splitting the image into blocks, and make it differentiable. In experiments, VAEs trained with the new loss function generated realistic, high-quality image samples. Compared to using the Euclidean distance and the Structural Similarity Index, the images were less blurry; compared to deep neural network based losses, the new approach required less computational resources and generated images with less artifacts.

## 1 Introduction

Variational Autoencoders (VAEs) [11] are generative neural networks that learn a probability distribution over $\mathcal{X}$ from training data $D = \{\mathbf{x}_0, ..., \mathbf{x}_n\} \subset \mathcal{X}$. New samples are generated by drawing a latent variable $\mathbf{z} \in \mathcal{Z}$ from a distribution $p(\mathbf{z})$ and using $\mathbf{z}$ to sample $\mathbf{x} \in \mathcal{X}$ from a conditional *decoder distribution* $p(\mathbf{x}|\mathbf{z})$. The distribution of $p(\mathbf{x}|\mathbf{z})$ induces a similarity measure on $\mathcal{X}$. A generic choice is a normal distribution $p(\mathbf{x}|\mathbf{z}) = \mathcal{N}(\mu_{\mathbf{x}}(\mathbf{z}), \sigma^2)$ with a fixed variance $\sigma^2$. In this case the underlying energy-function is $L(\mathbf{x}, \mathbf{x}') = \frac{1}{2\sigma^2}\|\mathbf{x} - \mathbf{x}'\|^2$. Thus, the model assumes that for two samples which are sufficiently close to each other (as measured by $\sigma^2$), the similarity measure can be well approximated by the squared loss. The choice of $L$ is crucial for the generative model. For image generation, traditional pixel-by-pixel loss metrics such as the squared loss are popular because of their simplicity, ease of use and efficiency [5]. However, they perform poorly at modeling the human perception of image similarity [30]. Most VAEs trained with such losses produce images that look blurred [3, 5]. Accordingly, perceptual loss functions for VAEs are an active research area. These loss functions fall into two broad categories, namely explicit models, as exemplified by the Structural Similarity Index Model (SSIM) [25], and learned models. The latter include models based on deep feature embeddings extracted from image classification networks [5, 30, 8] as well as combinations of VAEs with discriminator networks of Generative Adversarial Networks (GANs) [4, 13, 18].

Perceptual loss functions based on deep neural networks have produced promising results. However, features optimized for one task need not be a good choice for a different task. Our experimental results suggest that powerful metrics optimized on specific datasets may not generalize to broader categories of images. We argue that using features from networks pre-trained for image *classification* in loss functions for training VAEs for image *generation* may be problematic, because invariance properties beneficial for classification make it difficult to capture details required to generate realistic images.

---

Code and experiments are available at `github.com/SteffenCzolbe/PerceptualSimilarity`

| Reference | Dist. 1 | Dist. 2 | Reference | Dist. 1 | Dist. 2 | Reference | Dist. 1 | Dist. 2 |
|---|---|---|---|---|---|---|---|---|
| 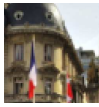 |  |  |  |  |  |  |  |  |

|  | Dist. 1 | Dist. 2 | | Dist. 1 | Dist. 2 | | Dist. 1 | Dist. 2 |
|---|---|---|---|---|---|---|---|---|
| Human | ✓ | | | ✓ | | | ✓ | |
| Watson-DFT | ✓ | | | ✓ | | | | ✗ |
| Baselines | ✗ | | | | ✗ | | | ✗ |
| Deep Models | ✗ | | | | ✗ | | ✓ | |

Figure 1: Similarity judgement of tested metrics on selected images (see 4.1 for details). The proposed Watson-DFT metric can model spatial variations, yet punishes image degradation through noise and graphic artifacts. Deep neural network based metrics pre-trained on classification tasks are invariant to the image quality, leading to more artifacts when employed in generation tasks. We refer to Supplement F for additional random examples.

In this work, we introduce a loss function based on Watson's visual perception model [27], an explicit perceptual model used in image compression and digital watermarking [15]. The model accounts for the perceptual phenomena of sensitivity, luminance masking, and contrast masking. It computes the loss as a weighted distance in frequency space based on a Discrete Cosine Transform (DCT). We optimize the Watson model for image generation by (i) replacing the DCT with the discrete Fourier Transform (DFT) to improve robustness against translational shifts, (ii) extending the model to color images, (iii) replacing the fixed grid in the block-wise computations by a randomized grid to avoid artifacts, and (iv) replacing the max operator to make the loss function differentiable. We trained the free parameters of our model and several competitors using human similarity judgement data ([30], see Figure 1 for examples). We applied the trained similarity measures to image generation of numerals and celebrity faces. The modified Watson model generalized well to the different image domains and resulted in imagery exhibiting less blur and far fewer artifacts compared to alternative approaches.

## 2 Background

In this section we briefly review variational autoencoders and Watson's perceptual model.

**Variational Autoencoders**  Samples from VAEs [11] are drawn from $p(\mathbf{x}) = \int p(\mathbf{x}|\mathbf{z})p(\mathbf{z})\,\mathrm{d}\mathbf{z}$, where $p(\mathbf{z})$ is a prior distribution that can be freely chosen and $p(\mathbf{x}|\mathbf{z})$ is typically modeled by a deep neural network. The model is trained using a variational lower bound on the likelihood

$$\log p(\mathbf{x}) \leq \mathbb{E}_{q(\mathbf{z}|\mathbf{x})}\left\{\log p(\mathbf{x}|\mathbf{z})\right\} - \beta\mathrm{KL}(q(\mathbf{z}|\mathbf{x})\|p(\mathbf{z})) \ , \tag{1}$$

where $q(\mathbf{z}|\mathbf{x})$ is an encoder function designed to approximate $p(\mathbf{z}|\mathbf{x})$ and $\beta$ is a scaling factor. We choose $p(\mathbf{z}) = \mathcal{N}(0, I)$ and $q(\mathbf{z}|\mathbf{x}) = \mathcal{N}(\mu_{\mathbf{z}}(\mathbf{x}), \Sigma_{\mathbf{z}}(\mathbf{x}))$, where the covariance matrix $\Sigma_{\mathbf{z}}(\mathbf{x})$ is restricted to be diagonal and both $\mu_{\mathbf{z}}$ and $\Sigma_{\mathbf{z}}(\mathbf{x})$ are modelled by deep neural networks.

**Loss functions for VAEs**  It is possible to incorporate a wide range of loss functions into VAE-training. If we choose $p(\mathbf{x}|\mathbf{z}) \propto \exp(-L(\mathbf{x}, \mu_{\mathbf{x}}(\mathbf{z}))$, where $\mu_{\mathbf{x}}$ is a neural network and we ensure that $L$ leads to a proper probability function, the first term of (1) becomes

$$\mathbb{E}_{q(\mathbf{z}|\mathbf{x})}\left\{\log p(\mathbf{x}|\mathbf{z})\right\} = -\mathbb{E}_{q(\mathbf{z}|\mathbf{x})}\left\{L(\mathbf{x}, \mu_{\mathbf{x}}(\mathbf{z}))\right\} + \mathrm{const} \ . \tag{2}$$

Choosing $L$ freely comes at the price that we typically lose the ability to sample from $p(\mathbf{x})$ directly. If the loss is a valid unnormalized log-probability, Markov Chain Monte Carlo methods can be applied. In most applications, however, it is assumed that $\mu_{\mathbf{x}}(\mathbf{z}), \mathbf{z} \sim p(\mathbf{z})$ is a good approximation of $p(\mathbf{x})$ and most articles present means instead of samples. Typical choices for $L$ are the squared loss $L_2(\mathbf{x}, \mathbf{x}') = \|\mathbf{x} - \mathbf{x}'\|^2$ and $p$-norms $L_p(\mathbf{x}, \mathbf{x}') = \|\mathbf{x} - \mathbf{x}'\|_p$. A generalization of $p$-norm based losses is the "General and Adaptive Robust Loss Function" [1], which we refer to as *Adaptive-Loss*. When used to train VAEs for image generation, the Adaptive-Loss is applied to 2D DCT transformations of entire images. Roughly speaking, it then adapts one shape parameter (similar to a $p$-value) and one scaling parameter per frequency during training, simultaneously learning a loss function and a

generative model. A common visual similarity metric based on image fidelity is given by Structured Similarity (SSIM) [25], which bases its calculation on the covariance of patches. We refer to section A in the supplementary material for a description of SSIM.

Another approach to define loss functions is to extract features using a deep neural network and to measure the differences between the features from original and reconstructed images [5]. In [5], it is proposed to consider the first five layers $\mathcal{L} = \{1, \ldots, 5\}$ of VGGNet [21]. In [30], different feature extraction networks, including AlexNet [12] and SqeezeNet [6], are tested. Furthermore, the metrics are improved by weighting each feature based on data from human perception experiments (see Section 4.1). With adaptive weights $\omega_{lc} \geq 0$ for each feature map, the resulting loss function reads

$$L_{\text{fcw}}(\mathbf{x}, \mathbf{x}') = \sum_{l \in \mathcal{L}} \frac{1}{H_l W_l} \sum_{h,w,c=1}^{H_l, W_l, C_l} \omega_{lc} (y_{hwc}^l - \hat{y}_{hwc}^l)^2 \ , \tag{3}$$

where $H_l$, $W_l$ and $C_l$ are the height, width and number of channels (feature maps) in layer $l$. The normalized $C_l$-dimensional feature vectors are denoted by $y_{hw}^l = \mathcal{F}_{hw}^l(\mathbf{x})/\|\mathcal{F}_{hw}^l(\mathbf{x})\|$ and $\hat{y}_{hw}^l = \mathcal{F}_{hw}^l(\mathbf{x}')/\|\mathcal{F}_{hw}^l(\mathbf{x}')\|$, where $\mathcal{F}_{hw}^l(\mathbf{x}) \in \mathbb{R}^{C_l}$ contains the features of image $\mathbf{x}$ in layer $l$ at spatial coordinates $h, w$ (see [30] for details).

**Watson's Perceptual Model**    Watson's perceptual model of the human visual system [27] describes an image as a composition of base images of different frequencies. It accounts for the perceptual impact of luminance masking, contrast masking, and sensitivity. Input images are first divided into $K$ disjoint blocks of $B \times B$ pixels, where $B = 8$. Each block is then transformed into frequency-space using the DCT. We denote the DCT coefficient $(i, j)$ of the $k$-th block by $\mathbf{C}_{ijk}$ for $1 \leq i, j \leq B$ and $1 \leq k \leq K$.

The Watson model computes the loss as weighted $p$-norm (typically $p = 4$) in frequency-space

$$D_{\text{Watson}}(\mathbf{C}, \mathbf{C}') = \sqrt[p]{\sum_{i,j,k=1}^{B,B,K} \left| \frac{\mathbf{C}_{ijk} - \mathbf{C}'_{ijk}}{\mathbf{S}_{ijk}} \right|^p} \ , \tag{4}$$

where $\mathbf{S} \in \mathbb{R}^{K \times B \times B}$ is derived from the DCT coefficients $\mathbf{C}$. The loss is not symmetric as $\mathbf{C}'$ does not influence $\mathbf{S}$. To compute $\mathbf{S}$, an image-independent sensitivity table $\mathbf{T} \in \mathbb{R}^{B \times B}$ is defined. It stores the sensitivity of the image to changes in its individual DCT components. The table is a function of a number of parameters, including the image resolution and the distance of an observer to the image. It can be chosen freely dependent on the application, a popular choice is given in [2]. Watson's model adjusts $\mathbf{T}$ for each block according to the block's luminance. The luminance-masked threshold $\mathbf{T}_{\mathbf{L}_{ijk}}$ is given by

$$\mathbf{T}_{\mathbf{L}_{ijk}} = T_{ij} \left( \frac{\mathbf{C}_{00k}}{\bar{\mathbf{C}}_{00}} \right)^\alpha \ , \tag{5}$$

where $\alpha$ is a constant with a suggested value of $0.649$, $\mathbf{C}_{00k}$ is the d.c. coefficient (average brightness) of the $k$-th block in the original image, and $\bar{\mathbf{C}}_{00}$ is the average luminance of the entire image. As a result, brighter regions of an image are less sensitive to changes.

Contrast masking accounts for the reduction in visibility of one image component by the presence of another. If a DCT frequency is strongly present, an absolute change in its coefficient is less perceptible compared to when the frequency is less pronounced. Contrast masking gives

$$\mathbf{S}_{ijk} = \max(\mathbf{T}_{\mathbf{L}_{ijk}}, |\mathbf{C}_{ijk}|^r \mathbf{T}_{\mathbf{L}_{ijk}}^{(1-r)}) \ , \tag{6}$$

where the constant $r \in [0, 1]$ has a suggested value of $0.7$.

## 3    Modified Watson's Perceptual Model

**A differentiable model**    To make the loss function differentiable we replace the maximization in the computation of $\mathbf{S}$ by a smooth-maximum function $\text{smax}(x_1, x_2, \ldots) = \frac{\sum_i x_i e^{x_i}}{\sum_j e^{x_j}}$ and the equation for $\mathbf{S}$ becomes

$$\tilde{\mathbf{S}}_{ijk} = \text{smax}(\mathbf{T}_{\mathbf{L}_{ijk}}, |\mathbf{C}_{ijk}|^r \mathbf{T}_{\mathbf{L}_{ijk}}^{(1-r)}) \ . \tag{7}$$

For numerical stability, we introduce a small constant $\epsilon = 10^{-10}$ and arrive at the trainable Watson-loss for the coefficients of a single channel

$$L_{\text{Watson}}(\mathbf{C}, \mathbf{C}') = \sqrt[p]{\epsilon + \sum_{i,j,k=1}^{B,B,K} \left| \frac{\mathbf{C}_{ijk} - \mathbf{C}'_{ijk}}{\tilde{\mathbf{S}}_{ijk}} \right|^p} \quad . \tag{8}$$

**Extension to color images** Watson's perceptual model is defined for a single channel (i.e., greyscale). To make the model applicable to color images, we aggregate the loss calculated on multiple separate channels to a single loss value.[1] We represent color images in the YCbCr format, consisting of the luminance channel Y and chroma channels Cb and Cr. We calculate the single-channel losses separately and weight the results. Let $L_Y$, $L_{\text{Cb}}$, $L_{\text{Cr}}$ be the loss values in the luminance, blue-difference and red-difference components for any greyscale loss function. Then the corresponding multi-channel loss $L$ is calculated as

$$L = \lambda_Y L_Y + \lambda_{\text{Cb}} L_{\text{Cb}} + \lambda_{\text{Cr}} L_{\text{Cr}} \quad , \tag{9}$$

where the weighting coefficients are learned from data, see below.

**Fourier transform** In order to be less sensitive to small translational shifts, we replace the DCT with a discrete Fourier Transform (DFT), which is in accordance with Watson's original work (e.g., [29, 26]). The later use of the DCT was most likely motivated by its application within JPEG [24, 28]. The DFT separates a signal into amplitude and phase information. Translation of an image affects phase, but not amplitude. We apply Watson's model on the amplitudes while we use the cosine-distance for changes in phase information. Let $\mathbf{A} \in \mathbb{R}^{B \times B}$ be the amplitudes of the DFT and let $\Phi \in \mathbb{R}^{B \times B}$ be the phase-information. We then obtain

$$L_{\text{Watson-DFT}}(\mathbf{A}, \Phi, \mathbf{A}', \Phi') = L_{\text{Watson}}(\mathbf{A}, \mathbf{A}') + \sum_{i,j,k=1}^{B,B,K} w_{ij} \arccos\left[\cos(\Phi_{ijk} - \Phi'_{ijk})\right] \quad , \tag{10}$$

where $w_{ij} > 0$ are individual weights of the phase-distances that can be learned (see below).

The change of representation going from DCT to DFT disentangles amplitude and phase information, but does not increase the number of parameters as the DFT of real images results in a Hermitian complex coefficient matrix (i.e., the element in row $i$ and column $j$ is the complex conjugate of the element in row $j$ and column $i$).

**Grid translation** Computing the loss from disjoint blocks works for the original application of Watson's perceptual model, lossy compression. However, a powerful generative model can take advantage of the static blocks, leading to noticeable artifacts at block boundaries. We solve this problem by randomly shifting the block-grid in the loss-computation during training. The offsets are drawn uniformly in the interval $[\![-4, 4]\!]$ in both dimensions. In expectation, this is equivalent to computing the loss via a sliding window as in SSIM.

**Free parameters** When benchmarking Watson's perceptual model with the suggested parameters on data from a Two-Alternative Forced-Choice (2AFC) task measuring human perception of image similarity, see Subsection 4.1, we found that the model underestimated differences in images with strong high-frequency components. This allows compression algorithms to improve compression ratios by omitting noisy image patterns, but does not model the full range of human perception and can be detrimental in image generation tasks, where the underestimation of errors in these frequencies might lead to the generation of an unnatural amount of noise. We solve this problem by training all parameters of all loss variants, including $p$, $\mathbf{T}$, $\alpha$, $r$, $w_{ij}$ and for color images $\lambda_Y$, $\lambda_{\text{Cb}}$ and $\lambda_{\text{Cr}}$, on the 2AFC dataset (see Section 4.1).

# 4 Experiments

We empirically compared our loss function to traditional baselines and the recently proposed Adaptive-Loss [1] as well as deep neural network based approaches [30]. First, we trained the free parameters of the proposed Watson model as well as of loss functions based on VGGNet [21] and SqueezeNet [6] to mimic human perception on data of human perceptual judgements. Next, we applied the similarity metrics as loss functions of VAEs in two image generation tasks. Finally, we evaluated the perceptual performance, and investigate individual error cases.

## 4.1 Training on data from human perceptual experiments

The modified Watson model, referred to as Watson-DFT, as well as LPIPS-VGG and LPIPS-Squeeze have tune-able parameters, which have to be chosen before use as a loss function. We train the parameters using the same data. For LPIPS-VGG and LPIPS-Squeeze, we followed the methodology called *LPIPS (linear)* in [30] and trained feature weights according to (3) for the first 5 or 7 layers, respectively.

We trained on the Two-Alternative Forced-Choice (2AFC) dataset of perceptual judgements published as part of the Berkeley-Adobe Perceptual Patch Similarity (BAPPS) dataset [30]. Participants were asked which of two distortions $\mathbf{x}_1, \mathbf{x}_2$ of an $64 \times 64$ color image $\mathbf{x}_0$ is more similar to the reference $\mathbf{x}_0$. A human reference judgement $p \in [0, 1]$ is provided indicating whether the human judges *on average* deemed $\mathbf{x}_1$ ($p < 0.5$) or $\mathbf{x}_2$ ($p > 0.5$) more similar to $\mathbf{x}_0$.[2] The dataset is based on a total of 20 different distortions, with the strength of each distortion randomized per sample. Some distortions can be combined, giving 308 combinations. Figure 1 and Fig. B.7 in the supplementary material show examples.

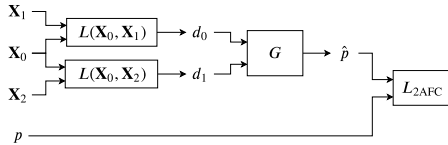

Figure 2: Optimization of a loss function $L$ on the 2AFC dataset. The inputs are the original image $\mathbf{x}_0$ and two distorted version $\mathbf{x}_1$ and $\mathbf{x}_2$. $L(\mathbf{x}_0, \cdot)$ is used to calculate perceptual distances $d_0, d_1$. The function $G$ predicts a ranking-probability $\hat{p}$. The training loss is the binary cross-entropy between the true-human ranking-probability $p$ and the loss-ranking-probability $\hat{p}$.

To train a loss function $L$ on the 2AFC dataset, we follow the schema outlined in Figure 2. We first compute the perceptual distances $d_0 = L(\mathbf{x}_0, \mathbf{x}_1)$ and $d_1 = L(\mathbf{x}_0, \mathbf{x}_2)$. Then these distances are converted into a probability to determine whether $(\mathbf{x}_0, \mathbf{x}_1)$ is perceptually more similar than $(\mathbf{x}_0, \mathbf{x}_2)$. To calculate the probability based on distance measures, we use

$$G(d_0, d_1) = \begin{cases} \frac{1}{2}, & \text{if } d_0 = d_1 = 0 \\ \sigma\left(\gamma \frac{d_1 - d_0}{|d_1| + |d_0|}\right), & \text{otherwise} \end{cases}, \tag{11}$$

where $\sigma(x)$ is the sigmoid function with learned weight $\gamma > 0$ modelling the steepness of the slope. This computation is invariant to linear transformations of the loss functions.

The training loss between the predicted judgment $G(d_0, d_1)$ and the human judgment $p$ is calculated by the binary cross-entropy:

$$L_{\text{2AFC}}(d_0, d_1) = p \log(G(d_0, d_1)) + (1 - p) \log(1 - G(d_0, d_1)) \tag{12}$$

This objective function was used to adapt the parameters of all considered metrics (used as loss functions in the VAE experiments). We trained the DCT based loss Watson-DCT and the DFT based loss Watson-DFT, see (8) and (10), respectively, both for single-channel greyscale input as well as for color images with the multi-channel aggregator (9). We compared our results to the linearly weighted deep loss functions from [30], which we reproduced using the original methodology, which differs from (3) only in modeling $G$ as a shallow neural network with all positive weights.



|  (a) Watson-DFT | (b) SSIM | (c) Adaptive-Loss | (d) LPIPS-Squeeze | (e) LPIPS-VGG |

Figure 3: Manifolds extracted from the 2-dimensional latent space of VAEs trained with different loss functions. Underlying $\mathbf{z}$-values lie on a grid over $\mathbf{z} \in [-1.5, 1.5]^2$.

## 4.2 Application to VAEs

We evaluated VAEs trained with our pre-trained modified Watson model, pre-trained deep-learning based LPIPS-VGG and LPIPS-Squeeze, and not pre-trained baselines SSIM and Adaptive-Loss. The latter adapted the parameters of the loss function during VAE training. We used the implementations provided by the original authors when available. Since quantitative evaluation of generative models is challenging [23], we qualitatively assessed the generation, reconstruction and latent-value interpolation of each model on two independent datasets.[3] We considered the gray-scale MNIST dataset [14] and the celebA dataset [16] of celebrity faces. The images of the celebA dataset are of higher resolution and visual complexity compared to MNIST. The feature space dimensionalities for the two models, MNIST-VAE and celebA-VAE, were 2 and 256, respectively.[4]

Results of reconstructed samples from models trained on celebA are given in Fig. 4. Generated images of all models are given in Fig. 5 and Supplement D. For the two-dimensional feature-space of the MNIST model, Fig. 3 shows reconstructions from $\mathbf{z}$-values that lie on a grid over $\mathbf{z} \in [-1.5, 1.5]^2$. Additional results showing interpolations and reconstructions of the models are given in Supplement D.

**Handwritten digits**    The VAE trained with the Watson-DFT captured the MNIST dataset well (see Fig. 3 and supplementary Fig. D.8). The visualization of the latent-space shows natural-looking handwritten digits. All generated samples are clearly identifiable as numbers. The models trained with SSIM and Adaptive-Loss produced similar results, but edges are slightly less sharp (Fig. D.8). The VAE trained with the LPIPS-VGG metric produced unnatural looking samples, very distinct from the original dataset. Samples generated by VAEs trained with LPIPS-Squeeze were not recognizeable as digits. Both deep feature based metrics performed badly on this simple task; they did not generalize to this domain of images, which differs from the 2AFC images used to tune the learned similarity metrics.

**Celebrity photos**    The model trained with the Watson-DFT metric generated samples of high visual fidelity. Background patterns and haircuts were defined and recognizable, and even strands of hair were partially visible. The images showed no blurring and few artifacts. However, objects lacked fine details like skin imperfections, leading to a smooth appearance. Samples from this generative model overall looked very good and covered the full range of diversity of the original dataset.

The VAE trained with SSIM showed the typical problems of training with traditional losses. Well-aligned components of the images, such as eyes and mouth, were realistically generated. More specific features such as the background and glasses, or features with a greater amount of spatial

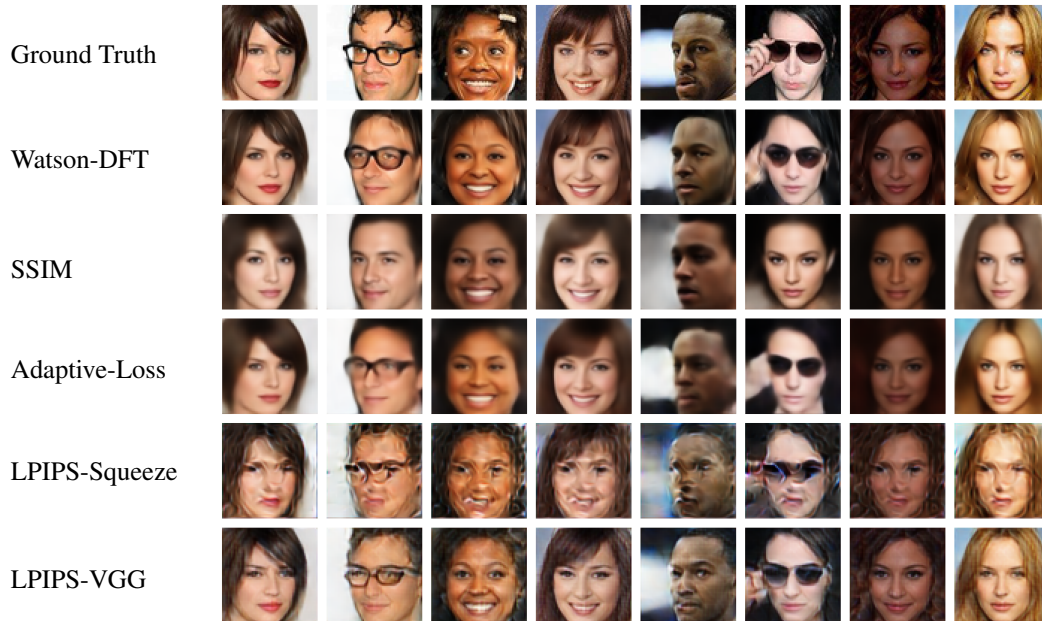

Figure 4: Reconstructions from the celebA test set using VAEs trained with different loss functions.

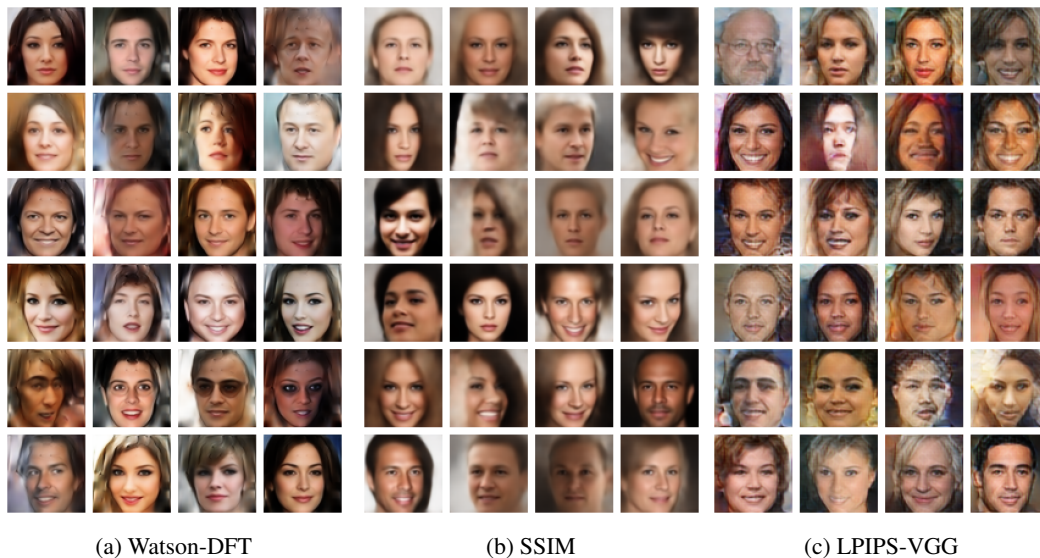

(a) Watson-DFT         (b) SSIM         (c) LPIPS-VGG

Figure 5: Random samples decoded from latent values $\mathbf{z} \sim P(\mathbf{z})$ for VAEs trained with different loss functions. For results of Adaptive-Loss and LPIPS-Squeeze, we refer to supplementary material Appendix D.

uncertainty, such as hair, were very blurry or not generated at all. The model trained with Adaptive-Loss improves on color accuracy, but blurring is still an issue. The VAE trained with the LPIPS-VGG metric generated samples and visual patterns of the original dataset very well. Minor details such as strands of hair, skin imperfections, and reflections were generated very accurately. However, very strong artifacts were present (e.g., in the form of grid-like patterns, see Fig. 5 (c)). The Adaptive-Loss gave similar results as SSIM, see supplementary Fig. D.11 (a). The VAE trained with LPIPS-Squeeze showed very strong artifacts in reconstructed images as well as generated images, see supplementary Fig. D.11 (b)).

Table 1: Time and GPU memory required for a typical learning scenario. We omit metrics without an explicit greyscale implementation from the greyscale test. Lower values are better.

| Input | Metric | Runtime (s) | Mem. (Mb) |
|---|---|---:|---:|
| Grey | $L_2$ | 0.3 | 8 |
| | SSIM | 5.5 | 38 |
| | Watson-DFT | 4.4 | 38 |
| Color | $L_2$ | 0.3 | 24 |
| | SSIM | 8.7 | 114 |
| | Adaptive-Loss | 8.7 | 132 |
| | Watson-DFT | 16.2 | 114 |
| | LPIPS-Squeeze | 14.1 | 546 |
| | LPIPS-VGG | 56.8 | 2214 |

### 4.3 Perceptual score

We used the validation part of the 2AFC dataset to compute perceptual scores and investigated similarity judgements on individual samples of the set. The agreement with human judgements is measured by $p\hat{p} + (1-p)(1-\hat{p})$ as in [30].[5] A human reference score was calculated using $p = \hat{p}$. The results are summarized in Figure 6. Overall, the scores were similar to the results in [30], which verifies our methodology. We can see that the explicit approaches ($L_2$ and SSIM) performed similarly. Adaptive-Loss, despite the ability ot adapt to the dataset, offers no improvement over the baselines. Watson-DFT performed considerably better, but not as well as LPIPS-VGG or LPIPS-Squeeze. We observe that the ability of metrics to learn perceptual judgement grows with the degrees of freedom (>1000 parameters for deep models, <150 for Watson-based metrics).

Inspecting the errors revealed qualitative difference between the metrics, some representative examples are shown in Fig. 1. We observed that the deep networks are good at semantic matching (see biker in Fig 1), but under-estimate the perceptual impact of graphical artifacts such as noise (see treeline) and blur. We argue that this is because the features were originally optimized for object recognition, where invariance against distortions and spatial shifts is beneficial. In contrast, the Watson-based metric is sensitive to changes in frequency (noise, blur) and large translations.

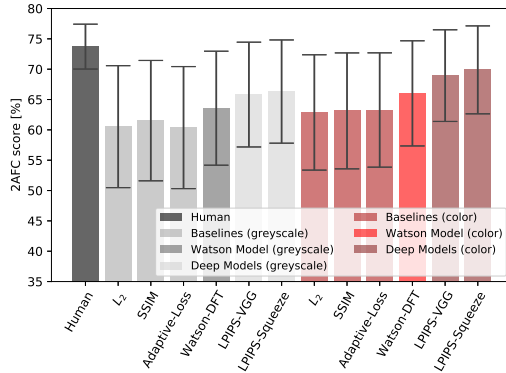

Figure 6: Metrics evaluated on the validation part of the 2AFC dataset (mean and variance). Black: Human reference. Grey: Metrics evaluated on greyscale images. Red: Metrics evaluated on color images. Shades group metrics into the categories of not pre-trained baseline metrics, our modified Watson Model and the deep-learning based LPIPS (linear) models from [30]. We refer to the supplementary material Appendix E for an evaluation by transformation group.

### 4.4 Resource requirements

During training, computing and back-propagating the loss requires computational resources, which are then unavailable for the VAE model and data. We measure the resource requirements in a typical learning scenario. Mini-batches of 128 images of size $64 \times 64$ with either one (greyscale) or three channels (color) were forward-fed through the tested loss functions. The loss with regard to one input image was back-propagated, and the image was updated accordingly using stochastic gradient descent. We measured the time for 500 iterations and the maximum GPU memory allocated. Results

are averaged over three runs of the experiment. Implementation in PyTorch [19], 32-bit precision, executed on a Nvidia Quadro P6000 GPU. The results are shown in Table 1. We observe that deep model based loss functions require considerably more computation time and GPU memory. For example, evaluation of Watson-DFT was 6 times faster than LPIPS-VGG and required only a few megabytes of GPU memory instead of two gigabytes.

# 5    Discussion and conclusions

**Discussion**    The 2AFC dataset is suitable to evaluate and tune perceptual similarity measures. But it considers a special, limited, partially artificial set of images and transformations. On the 2AFC task our metric based on Watson's perceptual model outperformed the simple $L_1$ and $L_2$ metrics as well as the popular structural similarity SSIM [25] and the Adaptive-Loss [1].

Learning a metric using deep neural networks on the 2AFC data gave better results on the corresponding test data. This does not come as a surprise given the high flexibility of this purely data-driven approach. However, the resulting neural networks did not work well when used as a loss function for training VAEs, indicating weak generalization beyond the images and transformations in the training data. This is in accordance with (1) the fact that the higher flexibility of LPIPS-Squeeze compared to LPIPS-VGG yields a better fit in the 2AFC task (see also [30]) but even worse results in the VAE experiments; (2) that deep model based approaches profit from extensive regularization, especially by including the squared error in the loss function (e.g., [8]). In contrast, our approach based on Watson's Perceptual Model is not very complex (in terms of degrees of freedom) and it has a strong inductive bias to match human perception. Therefore it extrapolates much better in a way expected from a perceptual metric/loss.

Deep neural networks for object recognition are trained to be invariant against translation, noise and blur, distortions, and other visual artifacts. We observed the invariance against noise and artifacts even after tuning on the data from human experiments, see Fig. 1. While these properties are important to perform well in many computer vision tasks, they are not desirable for image generation. The generator/decoder can exploit these areas of 'blindness' of the similarity metric, leading to significantly more visual artifacts in generated samples, as we observed in the image generation experiments.

Furthermore, the computational and memory requirements of neural network based loss functions are much higher compared to SSIM or Watson's model, to an extent that limits their applicability in generative neural network training.

In our experiments, the Adaptive-Loss, which is constructed of many similar components to Watson's perceptual model, did not perform much better than SSIM and considerably worse than Watson's model. This shows that our approach goes beyond computing a general weighted distance measure between images transformed to frequency space.

**Conclusion**    We introduced a novel image similarity metric and corresponding loss function based on Watson's perceptual model, which we transformed to a trainable model and extended to color-images. We replaced the underlying DCT by a DFT to disentangles amplitude and phase information in order to increase robustness against small shifts.

The novel loss function optimized on data from human experiments can be used to train deep generative neural networks to produce realistic looking, high-quality samples. It is fast to compute and requires little memory. The new perceptual loss function does not suffer from the blurring effects of traditional similarity metrics like Euclidean distance or SSIM, and generates less visual artifacts than current state-of-the-art losses based on deep neural networks.

# Acknowledgments and Disclosure of Funding

CI acknowledges support by the Villum Foundation through the project Deep Learning and Remote Sensing for Unlocking Global Ecosystem Resource Dynamics (DeReEco).

## Broader impact

The broader impact of our work is defined by the numerous applications of generative deep neural networks, for example the generation of realistic photographs and human faces, image-to-image translation with the special case of semantic-image-to-photo translation; face frontal view generation; generation of human poses; photograph editing, restoration and inpainting; and generation of super resolution images.

A risk of realistic image generation is of course the ability to produce "deepfakes". Generative neural networks can be used to replace a person in an existing image or video by someone else. While this technology has positive applications (e.g., in the movie industry and entertainment in general), it can be abused. We refer to a recent article by Kietzmann et al. for an overview discussing positive and negative aspects, including potential misuse that can affect almost anybody: "With such a powerful technology and the increasing number of images and videos of all of us on social media, anyone can become a target for online harassment, defamation, revenge porn, identity theft, and bullying — all through the use of deepfakes" [9].

We also refer to [9] for existing and potential commercial applications of deepfakes, such as software that allows consumers to "try on cosmetics, eyeglasses, hairstyles, or clothes virtually" and video game players to "insert their faces onto their favorite characters".

Our interest in generative neural networks, in particular variational autoencoders, is partially motivated by concrete applications in the analysis of remote sensing data. In a just started project, we will employ deep generative neural networks to the generation of geospatial data, which enables us to simulate the effect of human interaction w.r.t. ecosystems. The goal is to improve our understanding of these interactions, for example to analyse the influence of countermeasures such as afforestation in the context of climate change mitigation.

## Footnotes

[1]Many perceptually oriented image processing domains choose color representations that separate luminance from chroma. For example, the HSV color model distinguishes between hue, saturation, and color, and formats such as Lab or YCbCr distinguish between a luminance value and two color planes [22]. The separation of brightness from color information is motivated by a difference in perception. The luminance of an image has a larger influence on human perception than chromatic components [20]. Perceptual image processing standards such as JPEG compression utilize this by encoding chroma at a lower resolution than luminance [24].

[2]The three image patches $\mathbf{x}_0, \mathbf{x}_1, \mathbf{x}_2$ and label $p$ form a record. The dataset contains a total of 151,400 training records and 36,500 test records. Each training record was judged by 2, each test record by 5 humans.

[3]We provide the source code for our methods and the experiments, including the scripts that randomly sampled from the models to generate the plots in this article. We encourage to run the code and generate more samples to verify that the presented results are representative.

[4]The full architectures are given in supplementary material Appendix C. The optimization algorithm was Adam [10]. The initial learning rate was $10^{-4}$ and decreased exponentially throughout training by a factor of 2 every 100 epochs for the MNIST-VAE, and every 20 epochs for the celebA-VAE. For all models, we first performed a hyper-parameter search over the regularization parameter $\beta$ in (1). We tested $\beta = e^{\lambda}$ for $\lambda \in \mathbb{Z}$ for 50 epochs on the MNIST set and 10 epochs on the celebA set, then selected the best performing hyper-parameter by visual inspection of generated samples. Values selected for training the full model are shown in Table C.4 in the supplement. For each loss function, we trained the MNIST-VAE for 250 epochs and the celebA-VAE for 100 epochs.

[5]For example, when 80% of humans judged $x_1$ to be more similar to the reference we have $p = 0.2$. If the metric predicted $x_1$ to be closer, $\hat{p} = 0$, and we grant it 80% score for this judgement.

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
