[Supplementary Material]

# A Loss Function for Generative Neural Networks Based on Watson's Perceptual Model:
# Supplementary Material

## A  Structural Similarity Loss Function

The Structured Similarity (SSIM) [25], which models perceived image fidelity, is a popular loss function for VAE training. In SSIM, a sample is decomposed into blocks and individual channels. Errors are calculated per channel and finally averaged over the entire image. The structured similarity between two blocks $\mathbf{X}, \mathbf{Y} \in \mathbb{R}^{B \times B}$ is defined as

$$\text{SSIM}(\mathbf{X}, \mathbf{Y}) = \frac{(2m_{\mathbf{X}}m_{\mathbf{Y}} + c_1)(2\sigma_{\mathbf{XY}} + c_2)}{(m_{\mathbf{X}}^2 + m_{\mathbf{Y}}^2 + c_1)(\sigma_{\mathbf{X}}^2 + \sigma_{\mathbf{Y}}^2 + c_2)} \tag{A.13}$$

with $m_{\mathbf{X}}$ denoting the average of $\mathbf{X}$, $m_{\mathbf{Y}}$ the average of $\mathbf{Y}$, $\sigma_{\mathbf{X}}^2$ the variance of $\mathbf{X}$, $\sigma_{\mathbf{Y}}^2$ the variance of $\mathbf{Y}$ and $\sigma_{\mathbf{XY}}$ the co-variance of $\mathbf{X}$ and $\mathbf{Y}$. The constants $c_1 = (k_1 R)^2$ and $c_2 = (k_2 R)^2$ stabilize division and are calculated depending on the dynamic range $R$ of pixel values. We use the recommended values for the parameters $k_1 = 0.01$, $k_2 = 0.03$ and block size $B = 11$ [25]. Blocks are weighted by a Gaussian sampling function and moved pixel-by-pixel over the image.

## B  2AFC Data

Figure B.7: Example records from the 2AFC dataset. Top row: Original image patches. Row 2 & 3: Distortions. The distortion judged closer to the reference in human trials is marked red.

# C  Model Training

Table C.2: Architecture of the VAE for the MNIST dataset [14]. All convolutional layers use a stride of 1 and padding of 1. "Leaky ReLU" denotes leaky Rectified Linear Units [17]. Fully-connected layers state the number of hidden neurons.

| MNIST-VAE | Input Size | Layer |
|---|---|---|
| Encoder | $1 \times 32 \times 32$ | Conv. $3 \times 3$, leaky ReLU |
| | $32 \times 32 \times 32$ | Maxpool |
| | $32 \times 16 \times 16$ | Conv. $3 \times 3$, leaky ReLU |
| | $64 \times 16 \times 16$ | Fully-connected 1024, leaky ReLU |
| | 1024 | $2\times$ Fully-connected 2, leaky ReLU |
| Decoder | 2 | Fully-connected 1024, leaky ReLU |
| | 1024 | Fully-connected $64 \times 16 \times 16$, leaky ReLU |
| | $64 \times 16 \times 16$ | Conv. $3 \times 3$, leaky ReLU |
| | $64 \times 16 \times 16$ | Bilinear Upsampling |
| | $64 \times 32 \times 32$ | Conv. $3 \times 3$, leaky ReLU |
| | $32 \times 32 \times 32$ | Conv. $3 \times 3$, leaky ReLU |
| | $32 \times 32 \times 32$ | Conv. $3 \times 3$, Sigmoid |
| | $1 \times 32 \times 32$ | |

Table C.3: Architecture of the VAE for the celebA dataset [16]. All convolutional layers use a stride of 1 and padding of 1. "Leaky ReLU" denotes leaky Rectified Linear Units [17]. Fully-connected layers state the number of hidden neurons. We use batch normalization [7].

| celebA-VAE | Input Size | Layer |
|---|---|---|
| Encoder | $3 \times 64 \times 64$ | Conv. $3 \times 3$, leaky ReLU |
| | $64 \times 64 \times 64$ | Maxpool, Batch Normalization |
| | $64 \times 32 \times 32$ | Conv. $3 \times 3$, leaky ReLU |
| | $128 \times 32 \times 32$ | Maxpool, Batch Normalization |
| | $128 \times 16 \times 16$ | Conv. $3 \times 3$, leaky ReLU |
| | $128 \times 16 \times 16$ | Fully-connected 2048, leaky ReLU |
| | 2048 | $2\times$ Fully-connected 256, leaky ReLU |
| Decoder | 256 | Fully-connected 2048, leaky ReLU |
| | 2048 | Fully-connected $128 \times 16 \times 16$, leaky ReLU |
| | $128 \times 16 \times 16$ | Conv. $3 \times 3$, leaky ReLU |
| | $128 \times 16 \times 16$ | Bilinear Upsampling, Batch Normalization |
| | $128 \times 32 \times 32$ | Conv. $3 \times 3$, leaky ReLU |
| | $64 \times 32 \times 32$ | Bilinear Upsampling, Batch Normalization |
| | $64 \times 64 \times 64$ | Conv. $3 \times 3$, leaky ReLU |
| | $64 \times 64 \times 64$ | Conv. $3 \times 3$, leaky ReLU |
| | $64 \times 64 \times 64$ | Conv. $3 \times 3$, Sigmoid |
| | $3 \times 64 \times 64$ | |

Table C.4: Hyper-parameters for models trained.

| Model | Similarity Metric | Hyper-parameter $\beta$ |
|---|---|---|
| MNIST-VAE | Watson-DFT | $e^{-1}$ |
| | SSIM | $e^{-9}$ |
| | Adaptive-Loss | $e^{0}$ |
| | LPIPS-VGG | $e^{-9}$ |
| | LPIPS-Squeeze | $e^{-9}$ |
| celebA-VAE | Watson-DFT | $e^{1}$ |
| | SSIM | $e^{-12}$ |
| | Adaptive-Loss | $e^{-2}$ |
| | LPIPS-VGG | $e^{-10}$ |
| | LPIPS-Squeeze | $e^{-9}$ |

# D  Additional Results

Ground Truth

Watson-DFT

SSIM

Adaptive-Loss

LPIPS-Squeeze

LPIPS-VGG

Figure D.8: Reconstruction of samples from the MNIST test set using VAEs trained with different loss functions.

Ground Truth

Watson-DFT

SSIM

Adaptive-Loss

LPIPS-Squeeze

LPIPS-VGG

Figure D.9: Latent space interpolation between two samples from the MNIST test set. Comparison of VAEs trained with different loss functions.

Figure D.10: Latent space interpolation between two samples from the celebA test set. Comparison of VAEs trained with different loss functions.

(a) Adaptive-Loss        (b) LPIPS-Squeeze

Figure D.11: Random samples decoded from latent values $\mathbf{z} \sim P(\mathbf{z})$ for VAEs trained with Adaptive-Loss and LPIPS-Squeeze.

# E   Additional 2AFC Metrics

(a) Algorithms

(b) Distortions

Figure E.12: Metrics evaluated on transformation groups of the validation part of the 2AFC dataset (mean and variance). Transformations in (a) have been generated by established algorithms (Super-resolution, Frame Interpolation, Video Deblur, Colorization), transformations in (b) by distortions (Blur, Compression, Noise, CNN based distortions). For more details on data generation see [30].

# F Additional 2AFC Judgements

Figure F.13: Similarity judgements on the 2AFC dataset. First row: reference image. Second row: image judged more similar to reference by Watson-DFT metric. Third row: image judged more similar by LPIPS-VGG metric. Red framed: image judged more similar by 5 human judges. Images pictured were selected at random.