[Reviews · NeurIPS 2020]

Review 1

Summary and Contributions: The paper proposes to use an adapted version of Watson's Perceptual Model to train a VAE for higher perceptual quality than e.g. SSIM or a deep-feature based loss.

Strengths: Interesting approach, thorough description and motivation for the adaptions to Watson's model. I found the description of the components of Watson's model interesting.

Weaknesses: W1) The paper (rightfully) dismisses MSE in the introduction, but I would have liked to see a VAE trained with MSE as a comparison. It seems at least for MNIST, this should give somewhat reasonable models? If not, please elaborate why. W2) What's the difference between Deeploss-VGG/-Squeeze and the loss proposed in [29] (LPIPS)? As far as I know, they also use VGG and Squeeze. If it's the same, it would help understanding to call it the same. If not, a short note on what is different somewhere in L140-145 would be nice. W3) The paper only qualitatively evaluates the proposed method and "encourages [the reader] to run the code and generate more samples" (footnote 3), and on the 2AFC dataset (Fig 6). While I agree that there is no state-of-the-art perceptual quality metric, I would still have appreciated a discussion of candidates (e.g. FID / VMAF / NIQE / NIMA), and a plot. Minor: - L on L58 not defined, consider adding it to the previous sentence (...loss functions $L$ into...) - on L79, it would have helped to add a link to the further text, as I was unsure what the terms meant, but they are described (e.g. "luminance masking, contrast masking, and sensitivity *(described below)*")

Correctness: The approach seems correct.

Clarity: Fairly clearly written

Relation to Prior Work: The paper discussions some related work.

Reproducibility: Yes

Additional Feedback: This is not a weakness and more of a "future work", but personally I would have been interested in seeing the loss used for rate-constrained image compression, and a discussion of the relation to Michaeli and Blau's "Rate-Distortion-Perception trade-off" paper, which would argue that the proposed loss is still a distortion. Post-rebuttal update: Thanks for addressing my concerns. Overall, I'm still concerned about W3. While the shown results are nice, I would have preferred a more quantitative evaluation. However, I'm keeping my accept.


Review 2

Summary and Contributions: The paper proposes a reconstruction loss function that is based on a human perceptual model that is computed in the frequency space. The authors show that this model matches human judgements on different synthetic distortions. The authors also show that this loss can be used to train a VAE.

Strengths: Use of an appropriate image loss function is an important design choice in many generative modelling task, and the authors provide an explicitly defined (not learned) choice based on human perception. Such an approach may be particularly valuable in the design of generative models robust to adversarial noise. Additionally, finding good ways of evaluating generative models is important, as KID, FID and Inception Scores are not interpretable and are biased toward features relevant for classification. Having good image distances are important, and perhaps there is more inspiration to be found in a true perceptual approach.

Weaknesses: Unfortunately, the evidence does not show that this loss metric actually guides generative models towards better image generations, so the utility is limited. [Update after rebuttal: upon re-considering the results and reading the rebuttal, I retract my previous assessment, I do see that there are reasons for why warping artifacts could happen in some images. I agree that the image generations are good, and that the methodology is sound. However, I am not sure that the quantitative evaluation bears out that the proposed approach is superior, as the 2AFC derived metric is the only quantitative evaluation shown.]

Correctness: The VGG & SqueezeNet -optimized celebA images presented in this paper (fig 4 and 5) have strong warpingartifacts, which indicate that the model was not trained to convergence, as example generations shown in the "Deep Feature Consistent Variational Autoencoder" paper do not have such artifacts. (in theory the VGG-optimized results of this paper should represent the "Deep Feature Consistent Variational Autoencoder" paper). This calls into question the conclusions drawn. [Update after rebuttal: -- I no longer agree with what I wrote here, please see above update]

Clarity: Overall the paper is very clear and well written. The term "masking" is not very familiar to me, it would help to have an explanation of this word. In the experimental section, it was not stated that the celebA images were at 64x64 resolution. In the broader impact section, lines 302-308 are not relevant to the broader impact of this particular work. Additionally, the authors bring up "impact", but do not state what the impact is likely to be. I actually think that the perceptual loss may be able to help create adversarial robustness.

Relation to Prior Work: yes. It would be interesting to also have a discussion of the relationship of this work to perceptual image compression works.

Reproducibility: Yes

Additional Feedback: Experiments on MNIST seem contrived for deeploss metrics, as the statistics of MNIST digits are completely different from natural images. [Update after rebuttal and reviewing discussion - I do think that the paper has shown that using Watson's model makes sense for applications where deep supervision is unwanted, and it seems like one of the best choices for non-deep image similarity metrics]


Review 3

Summary and Contributions: This paper presents a simple perceptual loss function motivated by the same image processing basics as JPEG: a loss on (scaled) frequency-space representations of YCbCr image patches. This produces significantly better VAE reconstructions/samples than L2/SSIM loss on pixels, and also better results than "perceptual" losses based on deep features (LPIPS etc).

Strengths: I generally like this paper. Grounding the contribution in classic image processing and perceptual studies is satisfying, and the proposed model is well-motivated and simple. I think this direction of research is promising, as it seems that a lot of "deep" perceptual losses are needlessly complicated when the problem they address (that measuring the difference between images in terms of raw RGB pixel differences is problematic) is straightforward enough that a straightforward analytical alternative likely exists. The paper is well-written, and the claims in the paper appear to be well-validated empirically.

Weaknesses: I have one critical concern with this paper, which is that the proposed model presented here is extremely similar to one result from “A General and Adaptive Robust Loss Function”, Jonathan T. Barron, CVPR, 2019. Section 3.1 of that paper (going from the arxiv version) has results on improving reconstruction/sampling quality from VAEs by using a loss on DCT coefficients of YUV images, very similar to what is done here. They also propose a loss with a heavy-tailed distribution that looks a lot like Equation 8 of this submission, and present a method where they optimize over the scale of the loss being imposed on each coefficient of the DCT (similar to this submission). And the improvement in sample/reconstruction quality they demonstrate looks a lot like what is shown in this submission. Given these overwhelming similarities, I'm unable to support the acceptance of this paper without a comparison to the approach presented in that work. Another (less pressing) concern I had for this submission: I’m surprised and confused that the experiment in Figure 6 suggests that the Deeploss-* techniques are preferable to the proposed Watson-DFT technique, which (to my eye) seems to produce much better reconstructions and samples in Figures 4 & 5. What is the source of this mismatch? I trust my eyes more than I trust this benchmark, but I am reluctant to champion a paper that only has one empirical evaluation where the proposed technique is outperformed by such a significant margin. I agree with the claims in the text that there is value to the proposed model being simple and compact, but it is unfortunate that the only empirical result in the paper requires this defending. A user study (perhaps in the same format as the 2AFC/BAPPS dataset) run on the Celeb-A reconstructions or samples would be extremely helpful here. An ablation study of the proposed model components would also be helpful for understanding what aspects of the loss are contributing the most to its performance. This ties into my concerns about the lack of evaluation against the Baron CVPR 2019 paper, which seems like a strongly-ablated version of this proposed method.

Correctness: I didn't see any issues here.

Clarity: This paper is generally well-written. Here are some nits I observed while reading it: 11: “less artifacts” -> “fewer artifacts” It’s a little weird to train a VAE with image reconstruction loss that doesn’t correspond to a valid probability distribution over pixels, but the authors address this and how it removes the ability to draw samples from p(x). The authors say that MCMC can be used to draw samples in this context, but they do not elaborate or provide a citation for this. Some clarification here would be helpful. The $l$ notation in and around Equation 3 would be easier to parse if a different character ($\ell$?) were used instead. It was unclear to me exactly what free parameters the proposed Watson-DFT model has, is it just the table of DCT compression coefficients? Please clarify what aspects of this model are learned. Is Deeploss-VGG/Squeeze exactly the same as the commonly used LPIPS metric? The paper is a little unclear about this, as it says only that it is trained in the same way as [29]. I think most people reading this paper would want to see an evaluation against something that exactly corresponds to LPIPS or E-LPIPS. The text regarding the 2AFC dataset is a little unclear, as the text repeatedly uses “the 2AFC dataset” to refer to the 2AFC dataset in the BAPPS dataset from [29]. “2AFC” is a common name for a category of tests, so in the text the authors should be more explicit when referencing this particular dataset (just as one would refer to MNIST as “the classification dataset”, etc). Figure 6 would be easier to parse if gridlines were “on”.

Relation to Prior Work: In general the treatment of prior work was fine. I would have found the paper slightly easier to read if it rooted its prior work section more on the image processing / compression side, because (if the reader happens to know image compression) this makes the paper very easy to immediately understand: It's a VAE where the loss looks a lot like JPEG.

Reproducibility: Yes

Additional Feedback: I generally like this paper, and if it contained a comparison to the very similar CVPR 2019 paper I would be strongly in favor of acceptance. I hope the authors can provide a baseline evaluation in the rebuttal, at which point I will re-evaluate my assessment. POST REBUTTAL: I thank the authors for addressing my concerns about the related work, and am raising my score (under the assumption that the new comparison in the rebuttal will be used in the camera ready version of the paper).

[Author Response · NeurIPS 2020]

Thank you for your comments. Here we address the most important ones (we can and will also fix the others):

**Rev#1/Rev#4:** What's the difference between Deeploss-VGG/-Squeeze and the loss proposed in [29] (LPIPS)? It is the
same metric, as noted in L140. We wanted a consistent naming scheme in the paper, but see that this can be confusing.
We consider renaming it to LPIPS-VGG and LPIPS-Squeeze.

**Rev#1:** Comparison to MSE for VAE In our experience there are no big differences on MNIST between $L_2$ and MSE
and the SSIM shown (which is in general considered the superior loss for images). If any, the generated images look
slightly less crisp, however this effect is often dominated by the quality of hyper-parameter tuning.

Quantitative analysis on other score-functions We will provide a measure in the updated paper. The choice is a bit
difficult because, for example, an LPIPS based measure will work best on the models trained with a variant of LPIPS.

**Rev#2:** Evidence that loss actually guides generative models towards better image generations The proposed metric
does lead to *much* better results, see Section 4.2, figures 4, 5, and in the supplement Section E, figures E.8–E.11.

Correctness of results in relation to "Deep Feature Consistent Variational Autoencoder" In the article by Hou et al. the
weights are not trained but fixed to $\omega_{lc} = 1/C_l$ in our notation. When the weights are adapted based on another dataset,
the resulting losses will be different. This is especially true in our case. While the dataset used for tuning is relatively
large, it still covers only a small subset of relevant transformations and the number of tunable parameters is large. Thus,
overfitting to the dataset can introduce, e.g., warping artefacts.

**Rev#4:** Comparison to "A General and Adaptive Robust Loss Func-
tion", Jonathan T. Barron, CVPR, 2019. Thank you, we were not
aware of that interesting and relevant publication. There are many
difference to our approach: Barron performs a 2D DCT over the
entire image while we use a blockwise FFT, which allows us to also
consider phase differences. This improves perceptual accuracy sig-
nificantly compared to DCT. We weight the DCT/FFT frequencies
resulting in <200 weights. Barron learns a 'robustness' value for each
separate DCT frequency. Due to not using block-DCT, this results in
a lot of parameters (e.g., 49152 trainable parameters on $128 \times 128 \times 3$

Figure R.1: Reconstructions using VAE. 1$^{st}$ row: Ground truth; 2$^{nd}$: Watson-DFT, 3$^{rd}$: CVPR 2019 method (please zoom in).

images). We also use the YUV/YCbCr color space, but we weight the color channels by learning the importance of each,
while Barron weights them equally. Both approaches learn a "robustness" parameter determining the significance of
outliers ($\alpha$ vs. $p$). Barron learns this parameter during training on the generative task, but has to add some regularization
to make this work. We learn the perceptual parameters on a perceptual dataset, independent of the generative task
and no regularization is necessary. We conducted experiments using the CVPR 2019 method with the code thankfully
provided by the author, e.g., see figures R.1 and R.2 (we will add more to the paper, also showing generated images).
*The method performs well, but clearly worse than Watson-DFT.*

Performance differences in Figure-6 & user-study *This is an impor-*
*tant aspect/finding of our study, please revisit L240–L270.* Consider
what task is solved and measured in Figure 6: a dataset is generated
by applying a certain set of transformations to images. The test-set
is not an unbiased estimate of performance in a real application as it
does not consider all relevant transformations. Moreover, the gener-
ative task differs from the test-set insofar as the VAE, similar to the
adversary in a GAN, tends to find the weaknesses in the loss-function
in order to maximize the similarity of $q(z|x)$ to $N(0, I)$. A user-study
would be nice to have, but may not provide insights into the models.
We believe that the differences between the models are so large that
there is no need for a user-study to decide which images look better –

Figure R.2: Updated Figure 6 of the submission, now including $L_1$ and Watson-DCT and the CVPR 2019 method ("Adaptive").

we assume that all reviewers agree on the obvious visual differences (on random samples, more can be generated using
the software provided).

Free parameters & Ablation study of model components A full list of free parameters is given in L137. The maximum
number of parameters of our method is 135. Earlier during development we compared DFT and DCT within our model.
With DCT the model performed significantly worse in all tasks, on par with the other models. We included this result in
Fig. R.2 as "Watson-DCT".

MCMC and the loss as probability distribution We did not want to claim that one can use MCMC if the model is not a
valid probability distribution. We will clarify this part. When the loss is a valid unnormalized log-probability, we can
use standard MCMC techniques like HMC to sample from $p(x|z)$.

[Meta-Review · NeurIPS 2020]

Three knowledgeable referees support acceptance, and I also recommend acceptance. The key contribution of this submission is a new reconstruction loss for VAEs (somewhat like JPEG loss) that matches human perception more closely than traditional VAE reconstruction losses (e.g. negative Gaussian log likelihood). For applications where the goal is to generate sharp images rather than to maximize the likelihood of held-out data, the proposed method is a good alternative to other known ways of generating sharp images with VAEs (i.e, autoregressive/flow-based decoders and adversarial loss function). Unlike these alternatives, the proposed method introduces few additional parameters to learn from the data. R1's and R2's concern about the lack of quantitative measures of performance is justified, but the author response also makes a compelling point about the difficulty of picking a fair quantitative metric. Given aims of this submission (to generate images with VAEs that look realistic to people), the strong qualitative results presented seem adequate to validate the approach. The authors should consider revising the manuscript to address R4's question about how MCMC can be used to draw samples.